# Mechanical Properties of Chopped Basalt Fiber-Reinforced Lightweight Aggregate Concrete and Chopped Polyacrylonitrile Fiber Reinforced Lightweight Aggregate Concrete

**DOI:** 10.3390/ma13071715

**Published:** 2020-04-06

**Authors:** Yusheng Zeng, Xianyu Zhou, Aiping Tang, Peng Sun

**Affiliations:** 1School of Civil Engineering, Harbin Institute of Technology, Harbin 150090, China; zengyus@163.com (Y.Z.); daoyizxy@126.com (X.Z.); pengsun1992@126.com (P.S.); 2Key Lab of Structures Dynamic Behaviour and Control of the Ministry of Education, Harbin Institute of Technology, Harbin 150090, China

**Keywords:** lightweight aggregate concrete, oven-dried density, water absorption, mechanical properties, fiber reinforcement, calculation model.

## Abstract

In this study, an experimental investigation was conducted on the mechanical properties of lightweight aggregate concrete (LWAC) with different chopped fibers, including basalt fiber (BF) and polyacrylonitrile fiber (PANF). The LWAC performance was studied in regard to compressive strength, splitting tensile strength and shear strength at age of 28 days. In addition, the oven-dried density and water absorption were measured as well to confirm whether the specimens match the requirement of standard. In total, seven different mixture groups were designed and approximately 104 LWAC samples were tested. The test results showed that the oven-dried densities of the LWAC mixtures were in range of 1.819–1.844 t/m^3^ which satisfied the definition of LWAC by Chinese Standard. Additionally, water absorption decreased with the increasing of fiber content. The development tendency of the specific strength of LWAC was the same as that of the cube compressive strength. The addition of fibers had a significant effect on reducing water absorption. Adding BF and PANF into concrete had a relatively slight impact on the compressive strength but had an obvious effect on splitting tensile strength, flexural strength and shear strength enhancement, respectively. In that regard, a 1.5% fiber volume fraction of BF and PANF showed the maximum increase in strength. The use of BF and PANF could change the failure morphologies of splitting tensile and flexural destruction but almost had slight impact on the shear failure morphology. The strength enhancement parameter *β* was proposed to quantify the improvement effect of fibers on cube compressive strength, splitting tensile strength, flexural strength and shear strength, respectively. And the calculation results showed good agreement with test value.

## 1. Introduction

Lightweight aggregate concrete (LWAC) has become a main development focus in the construction field, owing to its low density, high strength and excellent durability [1,2,3]. Moreover, artificial lightweight aggregates, which are manufactured from industrial wastes, river silts and solid wastes, are important sources for green building and sustainable development [4,5,6]. As compared to normal strength concrete (NSC), the usage of LWAC can bring significant benefits. For example, reducing more than 20% of the weight of structures can reduce seismic loading [7], a lower elastic modulus can bring a longer period of natural vibration and better deformability [8], lower thermal conductivity and thermal expansion result in improved fire-resistance [9] and frost-resistance [10] and reducing the size of the members results in a lower cost of construction [11,12]. All of these advantages have led to LWAC being widely applied in super high-rise buildings, long-span bridges and marine structures [13,14]. 

However, LWAC shows a more pronounced brittle failure than NSC, this is a major problem in engineering applications [15]. Numerous research studies have shown that using fibers and polymers in concrete can effectively improve the strength-ductility [16,17,18]. Steel fiber is used most often, owing to its excellent environmental action-resistance and economic effects [19,20]. Li et al. investigated the flexural behavior of LWAC with steel fiber and the test results showed that the steel fiber could significantly improve the compressive and flexural strength of LWAC, as well as the post-cracking behavior [21]. Li et al. researched the shear performance of steel fiber-reinforced LWAC beams and reported that the shear-resistance capacity was enhanced by 25.1%, 35.9% and 43.6% with steel fiber amounts of 0.4%, 0.8% and 1.2%, respectively, as compared to those without fiber reinforcement [22]. Numerous studies showed the steel fiber’s effects on the mechanical properties had significantly improved. While, some defects of steel fiber had always been neglected, such as the reduction in workability, the increase in weight of the concrete (because of the high specific gravity) and the corrosion in water and salt solution. Additionally, different types of fibers for reinforcing LWACs show different responses on mechanical properties [23]. Basalt fiber (BF), which is produced from basalt rock, reveals excellent resistance to chemical and heat attacks [24,25,26] and has been proved to have enhanced mechanical properties of concrete, in the context of a reinforced composite [27,28]. Similarly, polyacrylonitrile fiber (PANF) is a synthetic fiber and specifically a polymer with a chain of carbon connected to one another; it is a hard, horny and high-melting material [29]. PANF shows excellent crack resistance and effective tensile and shear strength improvement, as well as enhancing the frost resistance of the concrete [30,31,32]. However, only a limited number of studies have focused on BF and PANF reinforced LWAC, especially for chopped fibers. 

In addition, previous research had confirmed that the strength of the lightweight aggregate was the main controlling factor for the strength of LWAC, particularly for high-strength LWAC [33,34]. Moreover, it is well-known that the mechanical properties of concrete will affect its usage in projects and the same applies to LWAC. Numerous experiments have been conducted to investigate the behaviors of LWAC under compressive load, splitting tensile load and flexural load, whereas relatively fewer studies have been devoted to addressing the shear strength, particularly in regard to the impacts on different types of chopped fibers. In addition, it is worth noting that multiple test methods have been suggested to measure the shear strength of concrete, such as the rectangular short beam direct shear method, single shear surface ‘Z’ shaped specimen method, four-point force gap-beam specimen method, four-point force beam with uniform depth and varying width method, biaxial tension-compression cubic specimen method and torsion thin-wall cylinder specimen method [35]. However, there is no uniform specification for the shear strength testing of concrete. According to reports form Zhang and Guo [36], the four-point force beam with uniform depth and varying width method was suggested over the other methods, owing to of sample test equipment requirements and its closeness in representing a pure shear stress state in the middle space of the specimen. Moreover, existing literatures of fiber-reinforced LWACs lack the quantifying of the improved effects of fibers on strengths, especially on shear strength, which will limit the application.

Above all, in this study, basalt fiber and polyacrylonitrile fiber are used to product LWAC specimens respectively. In total, seven different mixtures were designed, for investigating the effects of the fiber volume fraction on the mechanical properties of LWAC. The oven-density and water absorption of LWAC were also studied. Different sizes of samples were used in different tests, including the compressive strength test, splitting tensile strength test, flexural strength test and shear strength test. Moreover, the strength enhancement parameter *β* was proposed to describe the fiber enhancement effects on strengths and corresponding calculation models were proposed to predict the cube compressive strength, splitting tensile strength, flexural strength and shear strength, which providing the basis for design and application of fiber reinforced LWACs.

## 2. Materials and Methods

### 2.1. Raw Materials

This study used P∙O 42.5R Portland cement and employed silica fume and first grade fly ash as supplementary cementious materials. The chemical compositions and physical properties of the cement, silica fume and fly ash are listed in Table 1. A Lytag lightweight aggregate was used as the coarse aggregate in the LWAC and the geometries and fundamental properties are shown in Table 2. Two different high-performance fibers, that is, basalt fiber (BF) and polyacrylonitrile fiber (PANF), were used as reinforcement materials shown as Figure 1. The properties of the fibers are shown in Table 3. Natural river sand was used as the fine aggregate (maximum size = 4.0 mm, bulk density = 1516 kg/m^3^, fineness modulus = 2.5). A superplasticizer was used to improve the workability of all mixtures. It also should be noted that all the properties presented in Table 1, Table 2 and Table 3 are determined by manufacturers.

### 2.2. Mixture Proportion

According to the existing research and design methods for the mixture proportions of LWAC in the Chinese standard technical specification for LWAC (JGJ 51-2002) [37], the mixture properties of LWAC in this study are listed in Table 4. A water/binder ratio (w/b) of 0.29 is used to produce the LWAC. Seven mixtures were used to test the effects of BF and PANF and the volume fractions were 0.5%, 1% and 1.5%, respectively. Fly ash and silica fume were selected as supplementary cementitious materials, that is, fly ash with 12% replacement and silica fume with 8% replacement, respectively [26]. A superplasticizer (high-performance water-reducing agent) was used to produce samples as well. It should be noted that, the details of mixture names are explained as follows—“LY” represents Lytag lightweight aggregate; “B” and “P” represent BF and PANF respectively; number 0.5, 1 and 1.5 represent fiber volume fraction of 0.5%, 1% and 1.5% respectively.

### 2.3. Testing Methods

All of the mechanical properties, test methods and sizes of test samples are summarized in Figure 2 and follow the Chinese National Standards [37,38] (The unit in Figure 2 is mm). All LWAC testing samples were cured in a standard curing room maintained at 20 ± 2 °C and 95% ± 10% relative humidity until testing time. Then, 100 × 100 × 100 mm^3^ cubic samples were measured at the ages of 7, 14 and 28 days for cube compressive strength at a loading rate of 6 kN/s and at 28 days for the splitting tensile strength at a loading rate of 1 kN/s. The flexural strengths of 100 × 100 × 400 mm^3^ prism samples were measured at 28 days under four-point testing with displacement control, at a rate of 0.1 mm/min. In addition, 650 × 120 × 150 mm^3^ samples with uniform depth and varying-width were used to measure the shear strength at 28 days under four-point testing with displacement control, at a rate of 0.1 mm/min. All of the mechanical property tests were conducted using the WAW-100t universal testing machine. 100 × 100 × 100 mm^3^ cubic samples were dried to a constant weight for measuring the oven-dried density. In that regard, the definition of a constant weight for LWAC is a weight loss of less than 0.1%. Cubic samples with dimensions of 100 × 100 × 100 mm^3^ were used to test water absorption. Specimens were soaked in water after drying to a constant weight and measured the weight since soaking time reach 1, 3, 6, 12, 24 and 48 h, respectively.

## 3. Test Results and Discussion 

### 3.1. Oven-dried Density and Specific Strength

The oven-dried densities of all mixture groups are shown in Figure 3 and the groups using lightweight aggregates all had density values less than the critical value (1.950 t/m^3^) from the LWAC requirement form [37]. It is evident that adding fibers had a slight impact on the oven-dried density of the LWAC. The BF volume fraction increased from 0% to 1.5% and the oven-dried density of the LWAC increased from 1.808 to 1.850 t/m^3^, whereas the oven-dried density of the 1.5% PANF sample increased to 1.843 t/m^3^, indicating that fiber addition could increase the density of LWAC, which may be because that fiber could fill internal pore.

The specific strength is defined as the ratio of compressive strength at 28 days to the oven-dried density of the concrete and can reflect the high-strength characteristics of construction materials. The test results for the specific strengths of the LWAC mixtures are shown in Figure 4. By increasing the volume fraction of BF and PANF, the specific strength of LWAC increased as well. The specific strength of the plain LWAC sample is 31.4 MPa·m^3^/t and LWAC mixtures with 1.5% BF and PANF reinforcement increased to 35.2 and 34.0 MPa·m^3^/t, respectively. Based on the definition of specific strength, this increase phenomenon indicates that fiber play a more important role in crack-resistance than in filling in concrete. 

### 3.2. Water Absorption 

The water absorption values of the LWACs are shown in Figure 5. It is clear that the water absorption capacity of all mixture groups increased with an increase of soaking time. Within 6 h, water absorption soared and then followed by a slight increase in 6 to 48 h. It was also found that BF and PANF showed significant effects on water absorption reduction. As the BF volume fraction increased from 0–1.5%, the water absorption at 48 h decreased from 5.93% to 4.02%. The sample with 1.5% PANF showed the maximum effect on water absorption reduction as compared with that of a plain LWAC sample, as the 48 h water absorption decreased to 3.61%. In general, fibers in concrete will block capillary pores, resulting in decreased water absorption, Liu [39] showed the similar test results. According to Karahan [40], the water absorption is closely related to porosity and water absorption would decrease as the porosity reduces. Therefore, BF and PANF addition into LWACs The effect of fibers on hindering the absorption of water ingress has a significant meaning in regard to improving the durability of LWAC (especially for freeze-thaw resistance) and our research team continues to study this aspect.

### 3.3. Cube Compressive Strength

The influences of the cube compressive strength of the LWAC samples with fiber reinforcements (*f_cu_*) at 28 days are illustrated in Figure 6. It was found that adding BF and PANF could improve the cube compressive strength. But both 0.5% BF and 0.5% PANF showed a slight effect on cube compressive strength improvement, equivalent to only 5% and 1% higher than the plain LWAC, respectively. As for BF, when the volume fraction increased to 1% compressive strength achieved 63.7 MPa, equivalent to a 12% improvement. When the volume fraction of BF increased to 1.5%, the compressive strength was 64.3 MPa, which was 13% higher than that of the plain LWAC sample but compared with sample with 1% BF, the cube compressive strength only improving by 1%. Thus, it could be considered that the ascension effect was the same as in the sample with 1% BF. As volume fraction increase from 0.5% to 1.5%, cube compressive strength increased as well and cube compressive strength achieved a maximum value of 62.5 MPa, which was approximately 9% higher than that of the plain LWAC sample. Moreover, as combined with the discussion regarding oven-dried density. The mechanism of compressive strength enhancement from fiber reinforcement might be that, as the curing time continues, there is an enhancement of the bonding strength between the fiber and cement matrix. Another reason could be that LWACs have generally shown a higher shrinkage than NSC [41] and adding fibers into concrete could effectively restrain shrinkage development [42], thereby controlling the growth of cracks in sufficient time to improve the compressive strength. According to the test results, to obtain the best improvements in the compressive strength improvement of LWAC, the 1% BF and 1.5% PANF volume fractions are respectively suggested herein. The test results are shown in Table A1.

### 3.4. Splitting Tensile Strength 

As shown in Figure 7, it is clear that adding either BF or PANF has a positive effect on the splitting tensile strength (*f_st_*). With an increase of the fiber volume fraction, the splitting tensile strength increases. Increasing the BF volume from 0% to 1.5% resulted in the maximum improvement of the splitting tensile strength of the LWAC samples, achieving 5.00 MPa, equivalent to 28.2% higher than that of plain LWAC. PANF had a relatively lower promotion. By adding 0.5%, 1% and 1.5% volume fractions of PANF into LWAC, the splitting tensile strength values increased to 4.20 MPa, 4.49 MPa and 4.79 MPa, respectively, which were 7.7%, 15.1% and 22.8% higher than those of plain LWAC, respectively. The mechanisms by which the fiber effects provide improvements in splitting strength can be reflected directly using failure morphology, as shown in Figure 8. As for plain LWAC, there were no evident changes in the early loading process. When close to the peak load, a tiny vertical crack opened in the middle space of the sample and then with continued loading, the tiny crack quickly extended through the upper and lower surface, with evident concrete spalling out. Finally, a brittle failure occurred. However, the addition of fibers changed the failure morphologies of the LWAC. The early loading process caused vertical tiny cracks as well but with continued loading, the expansion and growth of cracks did not reduce the loading capacity and no evident spalling of concrete was found. The test results are shown in Table A1.

### 3.5. Flexural Strength 

The effects of BF and PANF on the flexural strengths of the LWAC mixtures are shown in Figure 9. As can be seen, both BF and PANF provided significant improvements in flexural strength. As the BF volume fraction increased from 0% to 1.5%, the flexural strength increased up to approximately 50%, reaching 5.37 MPa. In contrast, a 1.5% volume fraction of PANF results in an approximately 40% increase in the flexural strength, reaching 5.02 MPa. Although it is easy to see that an increase of the fiber volume fraction leads to an increase in flexural strength, the increase rates show some differences. When increasing the BF by an equal volume fraction, the increase rate of the flexural strength decreased from 0.5% to 1.5%. In contrast, for PANF, an increase of the equal volume fraction resulted in an approximately equal increase rate for the flexural strength. The mechanisms of the fiber enhancement of the flexural strength could be explained through the fiber prevent crevice theory, as proposed by Romualdi [43]. In this theory, fibers cross the crack and transfer the stress to the upper and lower surfaces of the crack when subjected to tensile loading and the stress concentration at the cracks is alleviated, so that the sample can continue to bear loads. Moreover, the distributions and orientations of fibers are decisive in the tensile strength of LWAC [44]. Therefore, fiber distributions and volume fractions are major factors for obtaining an optimum improvement/positive effect/enhancement of mechanical properties. Based on the test results, for chopped BF and PANF, 1.5% volume fractions of both BF and PANF were suggested for use in LWAC, to obtain the relative optimum improvement in flexural strength. The test results are shown in Table A1.

The failure morphologies of LWAC mixtures subjected to flexural loading are shown in Figure 10 and the LWAC sample with PANF reinforced showed a similar flexural failure morphology with plain LWAC sample—in the early loading process, there is no detectable cracks were observed. As close to the peak flexural load, a vertical crack in middle space grew quickly and crossed the section and then sample failure when reached ultimate load. In contrast, BF addition not only improved the peak flexural loading but also delayed the growth of the crack and improved toughness effectively compared PANF, which because of the fiber prevented the crevice effect. 

### 3.6. Shear Strength 

Figure 11 shows the shear failure morphology of the LWAC mixtures. As adding PANF and BF led to similar failure morphologies of the plain sample. When the loading reached approximately 80% of the shear capacity bearing, a crack occurred in the shear front. With loading continued, as the stress approached the ultimate shear stress, a main oblique crack occurred in the shear front and then the crack quickly spread to the upper and lower surfaces. When the loading reached the peak shear load, the samples were destroyed immediately and were cut into two parts. In general, there was only one oblique crack and the fracture interface was clear and tidy. In this regard, fiber addition could not change this failure phenomenon. 

The effects of BF and PANF on the shear strengths of the LWAC mixtures are shown in Figure 12. It can be seen that an increase in the volume fraction of either BF or PANF results in an increase of the shear strength of the LWAC samples. The volume fraction of the BF samples increased from to 0.5%, 1% and 1.5% and the shear strength increased to 3.14, 3.48 and 3.64 MPa, respectively, equal to 12%, 24% and 30% higher than that of the plain LWAC sample, respectively. In contrast, PANF showed a relatively smaller increasing effect on shear strength enhancement than BF. As the volume fraction of PANF increased to 0.5%, 1% and 1.5%, the shear strength increased to 3.06, 3.24, 3.50 MPa, respectively, equivalent to 9%, 16% and 25% higher than that of the plain LWAC sample, respectively. The test results are shown in Table A1.

## 4. Strength Calculation Method

### 4.1. Calculation Method for Cube Compressive Strength

The compressive strength of concrete is the basic strength index in concrete structural design and monitoring. In this study, to consider the fiber improvement effects on the compressive strength of LWACs, an enhancement parameter βcu was proposed, as follows:(1)$$f_{cu,f\;}{=f}_{cu}{\times \;\beta }_{cu}$$
where, *f_cu,f_* and *f_cu_* are the cube compressive strengths of the fiber-reinforced LWAC and plain LWAC, respectively. The failure morphology of the concrete under a compression load could be considered as a tension destruction owing to the transverse expansion. Thus, fibers can be used in the concrete to improve the compressive strength, by improving the tension behavior. Therefore, *β_cu_* could be rewritten as Equation (2):(2)$$\beta_{cu}=\alpha_{cu}\cdot v\cdot \sqrt{\frac{f_{t}{}_{,f}}{f_{t}}}+1.$$

Herein, *v* (%) is the fiber volume fraction; $\sqrt{f_{t,f}/f_{t}}$ is the strength ratio, representing the increase ratio of the tensile strength of the plain LWAC owing to fiber addition, where $f_{t,f}$ is the tensile strength of the fiber and $f_{t}$ is the tensile strength of the plain LWAC (which can be calculated by $f_{t}=0.26f_{cu}^{2/3}$[45]); and $\alpha_{cu}$ is a strength correction coefficient (used to reduce the error and similarly hereinafter). The final fitting equation is shown in Equation (3) and the fitting effect is shown in Figure 13. The results for *β_cu,c_* using Equation (3) and a comparison with the test results for *β_cu,t_* are listed in Table 5. The *R*^2^ value of the fitting was 0.9136 and the *β_cu,t_*/*β_cu,c_* range grew from 0.98 to 1.02, showing good agreement with the test results. Moreover, data from References [23] and [46] are used to validate the Equation (3) and the *β_cu,t_*/*β_cu,c_* value range grew from 0.99 to 1.01.
(3)$$\beta_{cu}=0.004\cdot v\cdot \sqrt{\frac{f_{t}{}_{,f}}{f_{t}}}+1$$

### 4.2. Calculation Models for Splitting Tensile Strength

Similarly, to reflect the effects of BF and PANF on the splitting tensile strength of LWAC, an enhancement parameter *β_st_* was proposed, as shown in Equation (4):(4)$$f_{st,f}=f_{st}\cdot \beta_{st},$$
where, $f_{st,f}$ is the splitting tensile strength of LWAC with fiber reinforcement; $f_{st}$ is the splitting tensile strength of plain LWAC; and *β_st_* is the enhancement coefficient of the splitting tensile strength. In general, adding fibers to concrete would directly impact the tensile behavior. Therefore, in this study, the fiber volume fraction *v* (%) and the increase ratio of the splitting tensile strength $\sqrt{f_{t,f}/f_{st}}$ were considered as variables. The fitting equation is shown in Equation (5) and the fitting effect is shown in Figure 14. The results for *β_st,c_* using Equation (5) and a comparison with the test results for *β_st,t_*, are listed in Table 6. The *R*^2^ value of the fitting is 0.911 and the *β_st,t_*/*β_st,c_* values range from 0.98–1.03, showing good agreement with the test results. Moreover, data from References [23] and [47] are used to validate the Equation (5) and the *β_st,t_*/*β_st,c_* value range grew from 1.02 to 1.14.
(5)$$\beta_{st}=0.009\cdot v\cdot \sqrt{\frac{f_{t}{}_{,f}}{f_{st}}}+1.$$

### 4.3. Calculation Models for Flexural Strength

A flexural strength enhancement parameter was proposed, as follows:(6)$$f_{f,f}=f_{f}\cdot \beta_{f}.$$

Herein, $f_{f,f}$ is the flexural strength of LWAC with fiber reinforcement and $f_{f}$ is the flexural strength of plain LWAC. As usual, adding fiber into the concrete enhanced the flexural strength, owing to the significant tensile behavior of the fibers. Therefore, in this study, the fiber volume fraction *v* (%) and the increase ratio of the flexural strength as influenced by the fiber tensile strength $\sqrt{f_{t,f}/f_{f}}$ were considered as variables and the fitting equation is shown in Equation (7), it should be noted that the constant 0.23, is the correction coefficient of effect of *v* on flexural strength enhancement. The fitting effect is shown in Figure 15. The results for *β_f,c_* using Equation (7) and a comparison with the test results for *β_f,t_* are shown in Table 7. The *R*^2^ value of the fitting is 0.947 and the *β_f,t_*/*β_f,c_* value ranges from 0.97–1.03, showing good agreement with the test results. Moreover, data from References [23] and [48] are used to validate the Equation (7) and the *β_f,t_*/*β_f,c_* value range grew from 0.98 to 1.09.
(7)$$\beta_{f}=0.006\cdot \sqrt{\frac{f_{t,f}}{f_{f}}}+0.23\cdot v+1.$$

### 4.4. Calculation Models for Shear Strength

A shear strength enhancement parameter was proposed to reflect the fiber effect on the shear strengths of the LWAC samples and is shown in Equation (8):(8)$$f_{s,f}=f_{s}\cdot \beta_{s}.$$

According to a previous study [36], in regards to a shear strength experiment for concrete with uniform depth and varying width, the shear stress of the shear area in the middle space of the sample is in a uniform distribution and the principal tensile stress $\sigma_{y}\leq 0.1\bar{\tau }$ and principal compressive stress $\sigma_{x}\leq 0.2\bar{\tau }$; it could be considered as a pure shear state. Therefore, in this study, it is considered that the fibers enhance shear strength by controlling growth of cracks, which is related to the volume fraction *v* (%) and fiber length *l* (mm). The tensile strength of the fiber can be neglected and the fitting equation is shown in Equation (9), herein, constant 0.117 is the correction coefficient of effect of *v* on shear strength enhancement and constant 0.016 is the correction coefficient of coupling effect of *v* and *l* on shear strength enhancement. The fitting effect is shown in Figure 16. The results for *β_s,c_* using Equation (9) and a comparison with the test results for *β_s,t_* are listed in Table 8. The *R*^2^ value of the fitting is 0.951 and the *β_f,t_*/*β_f,c_* value ranges from 0.98–1.02, showing good agreement with the test results. Moreover, data from References [46] and [49] are used to validate the Equation (9) and the *β_s,t_*/*β_s,c_* value range grew from 0.91 to 1.07.
(9)$$\beta_{s}=0.117\cdot v+0.016\cdot v\cdot l+1.$$

## 5. Conclusions

In this study, the mechanical properties of various LWAC mixtures with two types of fiber reinforcement were investigated. In total, the effects of fiber on the cube compressive strength, splitting tensile strength, flexural strength and shear strength were compared and analyzed. In addition, the failure morphologies of LWAC samples subjected to different loading types were studied to explain the development trends of the strengths. Then, the fiber enhancement parameter *β* was used to calculate the cube compressive strength, splitting tensile strength, flexural strength and shear strength of LWAC with fiber reinforcement. According to research results mentioned above, the following conclusions can be drawn.

Both BF and PANF have effects on the improvement of oven-dried density and the reduction of water absorption with the increase of volume fraction due to the role of filling internal pores. The specific strength showed a similar change tendency with cube compressive strength, which increases with the increase of the fiber content respectively, indicating that fiber play a more important role in crack-resistance than in filling in concrete.The addition of BF and PANF into LWAC samples showed a slight influence on the cube compressive strength but had significant effects on the splitting tensile strength, flexural strength and shear strength improvement, relatively. The flexural strength was the most sensitive to fiber addition. In that regard, synthesizing the fiber effect on strength enhancement, a 1.5% volume fraction of BF or PANF is suggested to obtain the optimal improvement in mechanical properties.BF and PANF had effects on the failure morphology changes of splitting tensile and flexural destruction but had almost no effect on the shear failure morphology.The strength enhancement parameters *β* can describe the fiber effects on strength enhancement of LWACs when subjected to different loading types. Moreover, the strength calculation Equations (3), (5), (7) and (9) were established and the results of calculation agree well with the testing results.

## Figures and Tables

**Figure 1 materials-13-01715-f001:**
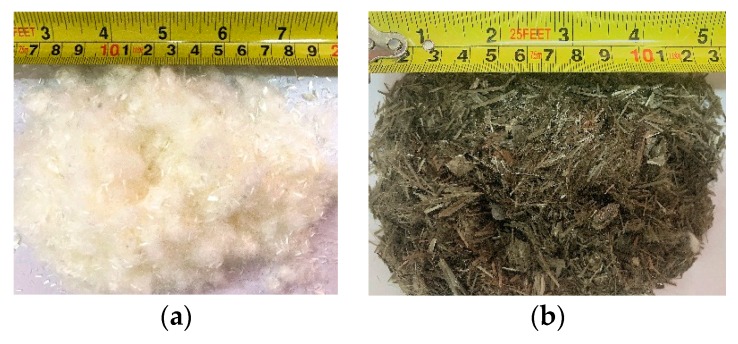
Chopped fiber features: (**a**) polyacrylonitrile fiber (PANF); (**b**) basalt fiber (BF).

**Figure 2 materials-13-01715-f002:**
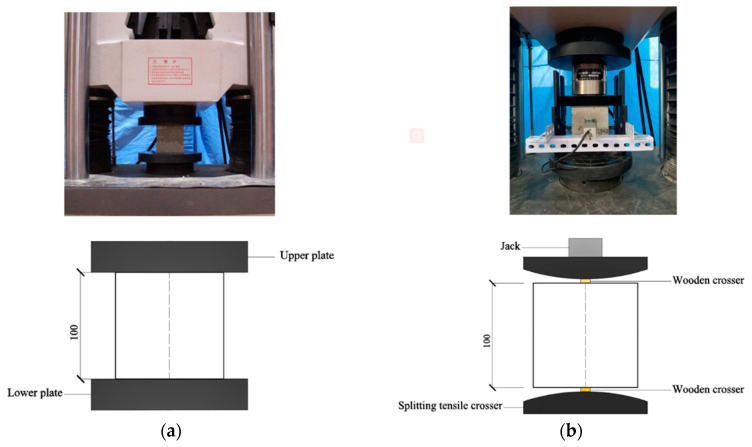
Test methods and size of test samples for—(**a**) cube compressive strength, (**b**) splitting tensile strength, (**c**) flexural strength, (**d**) shear strength. (The unit value of four figures are mm)

**Figure 3 materials-13-01715-f003:**
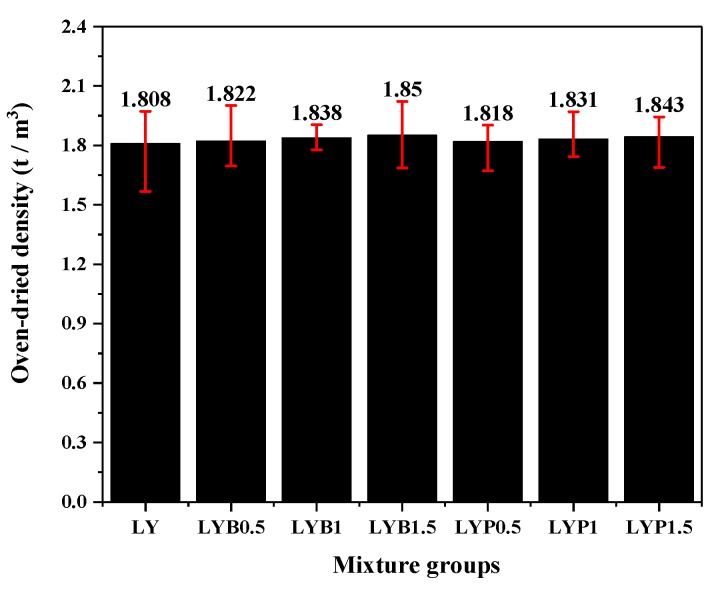
Oven-dried density of mixture groups.

**Figure 4 materials-13-01715-f004:**
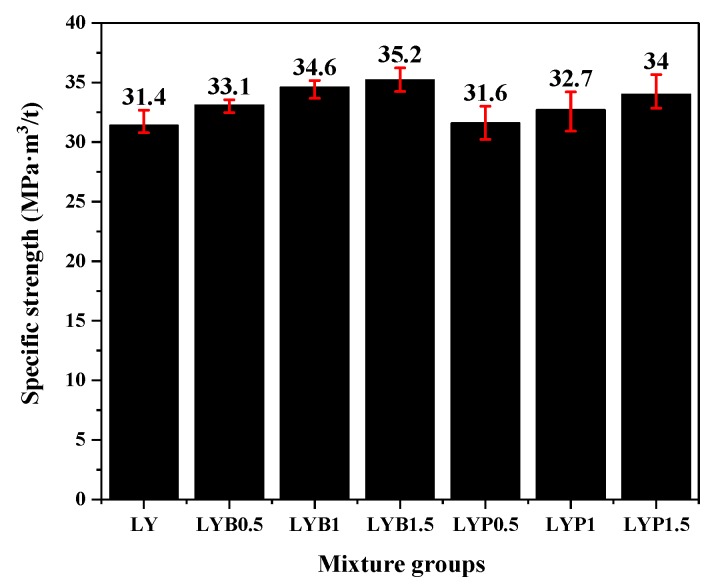
Specific strength of mixture groups.

**Figure 5 materials-13-01715-f005:**
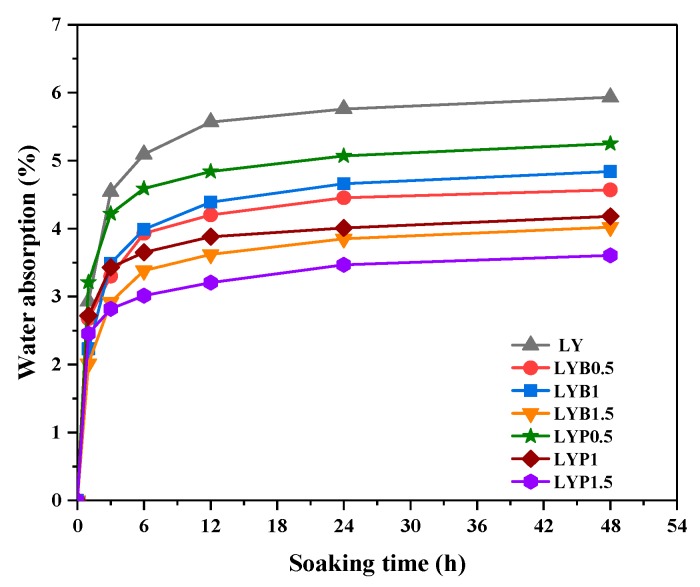
Water absorption of mixture groups.

**Figure 6 materials-13-01715-f006:**
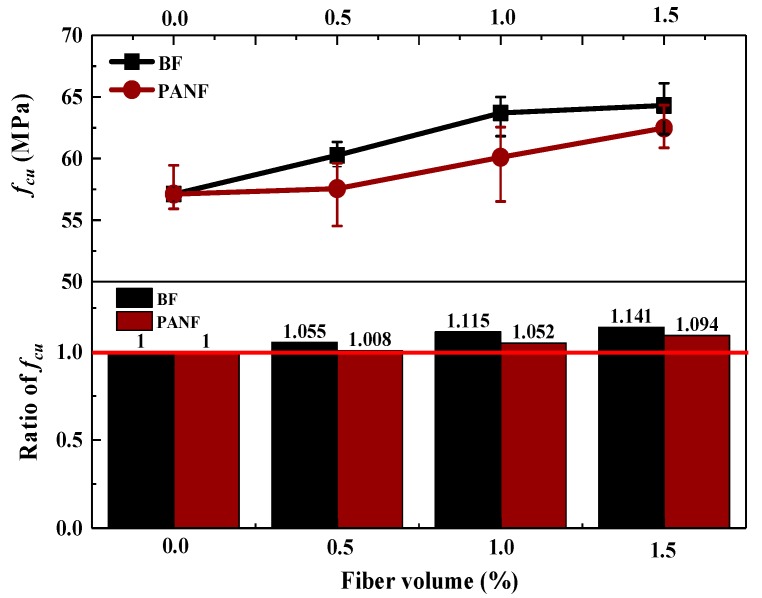
Effect of fibers on cube compressive strength of LWACs.

**Figure 7 materials-13-01715-f007:**
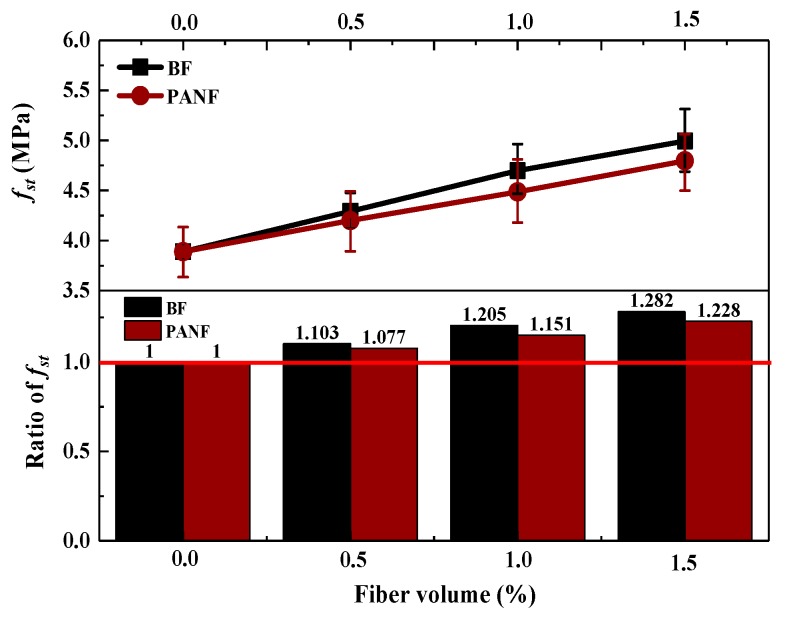
Effect of BF and PANF on splitting tensile strength of LWACs.

**Figure 8 materials-13-01715-f008:**
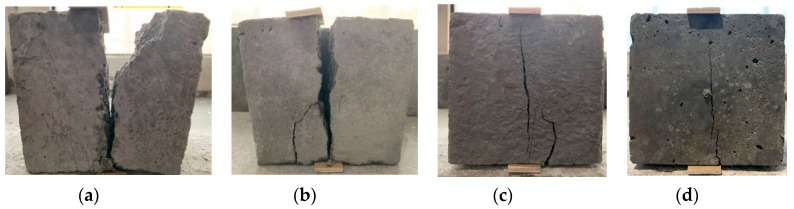
The splitting tensile failure morphology of LWACs—(**a**) plain sample, (**b**) 0.5% BF, (**c**) 1% BF, (**d**) 1.5% BF.

**Figure 9 materials-13-01715-f009:**
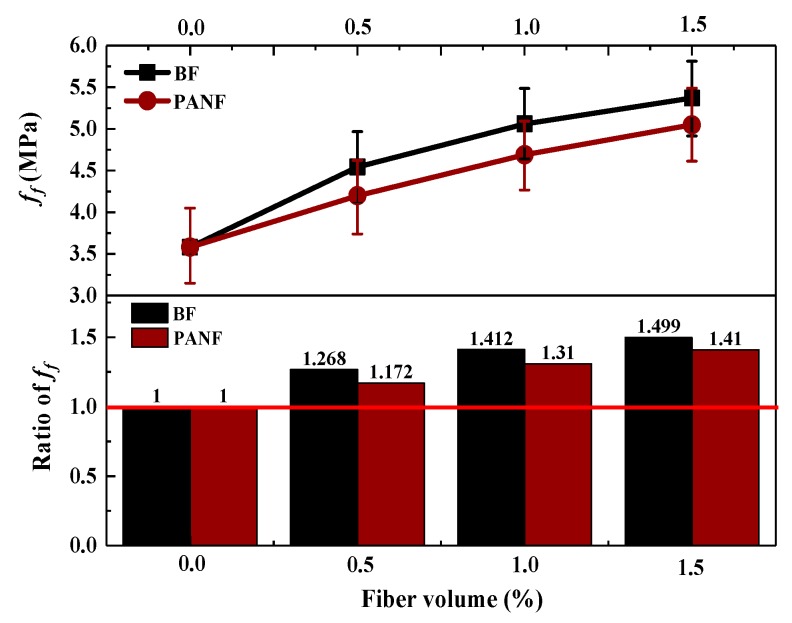
Effect of fibers on flexural strength of LWACs.

**Figure 10 materials-13-01715-f010:**
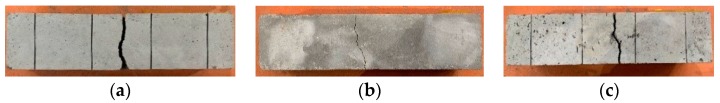
The flexural failure morphology of LWACs: (**a**) plain sample, (**b**) sample with BF, (**c**) sample with PANF.

**Figure 11 materials-13-01715-f011:**
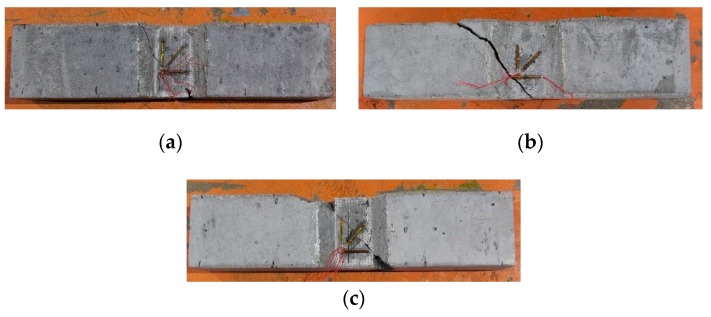
The shear failure morphology of LWACs: (**a**) plain sample, (**b**) sample with BF, (**c**) sample with PANF.

**Figure 12 materials-13-01715-f012:**
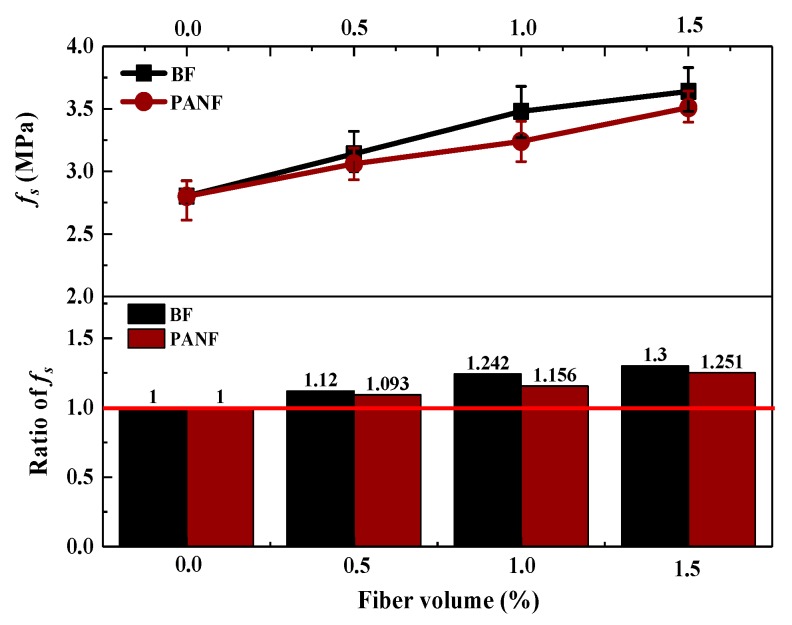
Effect of BF and PANF on shear strength of LWACs.

**Figure 13 materials-13-01715-f013:**
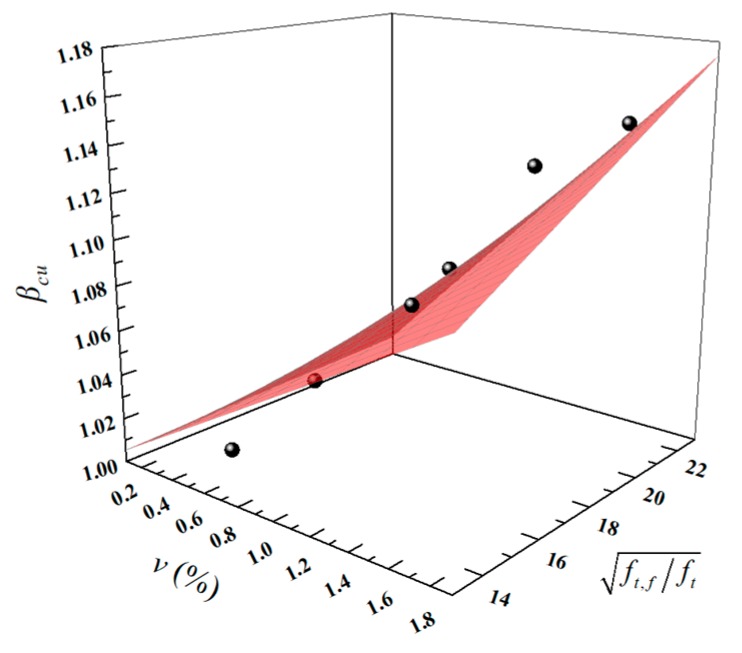
The fitting effect of *β_cu_*.

**Figure 14 materials-13-01715-f014:**
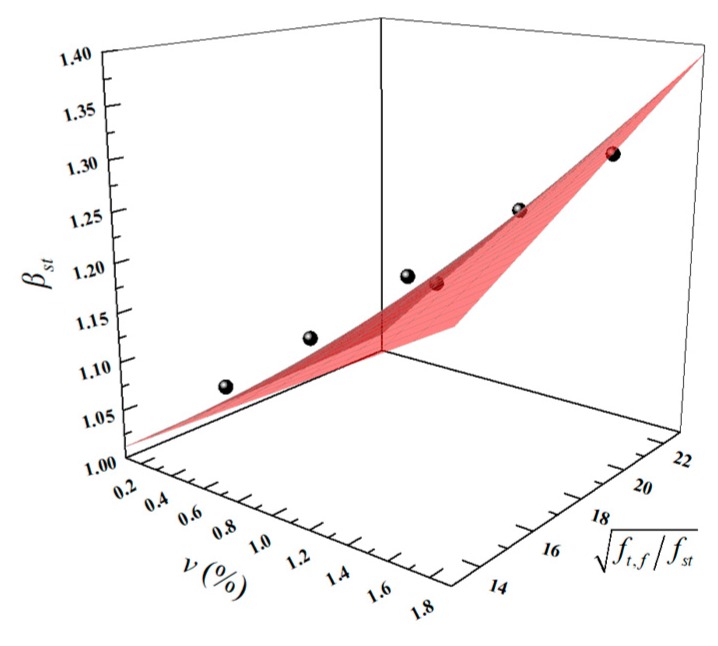
The fitting effect of *β_st_*.

**Figure 15 materials-13-01715-f015:**
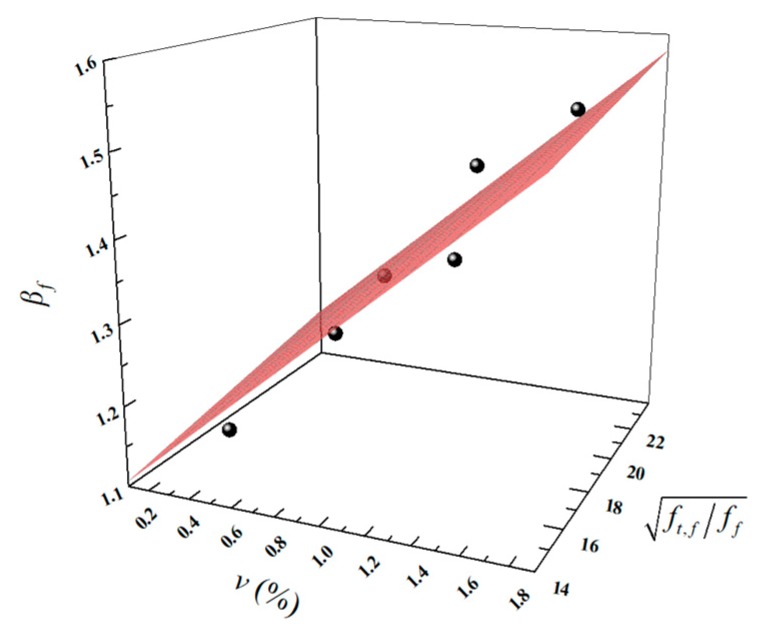
The fitting effect of *β_f_*.

**Figure 16 materials-13-01715-f016:**
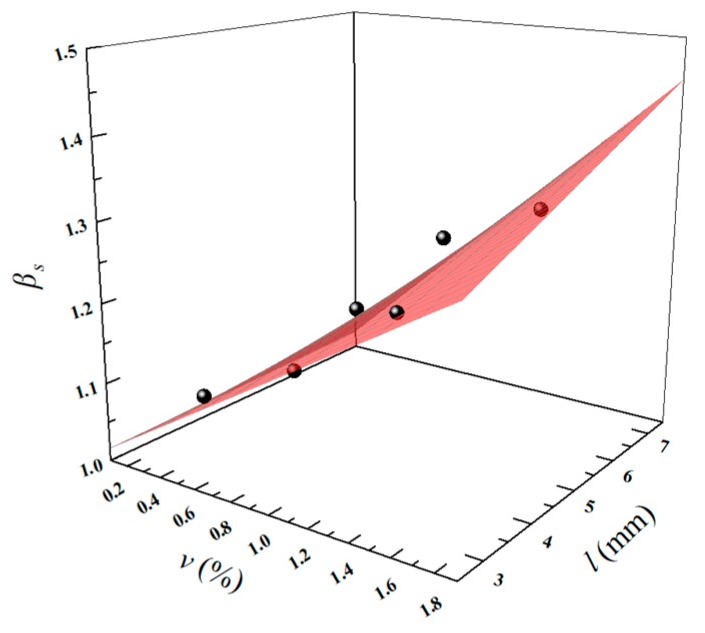
The fitting effect of *β_s_*.

**Table 1 materials-13-01715-t001:** The chemical compositions and physical properties of cement, silica fume and fly ash.

Materials	Density (kg/m^3^)	Mean Diameter (μm)	Loss on Ignition (%)	SiO_2_ (%)	CaO (%)	Al_2_O_3_ (%)	Fe_2_O_3_ (%)	MgO (%)
Cement	2234	18–24	-	22.1	62.39	5.72	3.05	2.02
Fly ash	2420	23–25	1.5	55.0	1.2	34.2	5.1	1.3
Silica fume	1700	0.1–0.3	3	95.0	0.2	0.7	0.6	0.5

**Table 2 materials-13-01715-t002:** The fundamental properties of coarse lightweight aggregate.

Types of Aggregates	Size (mm)	Bulk Density (kg/m^3^)	Cylinder Compressive Strength (MPa)	1h Water Absorption (%)	Softening Coefficient
Lytag	5-10	877	12.1	8.1	0.9

**Table 3 materials-13-01715-t003:** The fundamental properties of basalt fiber and polyacrylonitrile fiber.

Types of Fibers	Length (mm)	Density (kg/m^3^)	Tensile Strength (MPa)	Elongation(%)	Melting Point (°C)	Diameter (mm)	Elastic Modulus (GPa)
BF	6	2699	2000	2.5	1450	17.4	85
PANF	3	1190	800	1	245	10	≥10

**Table 4 materials-13-01715-t004:** Mixture proportions of lightweight aggregate concrete (LWAC) samples.

Groups	Cement (kg/m^3^)	Fly ash (kg/m^3^)	Silica Fume (kg/m^3^)	Coarse Aggregate (kg/m^3^)	Fine Aggregate (kg/m^3^)	Volume (%)	Super (kg/m^3^)	Water (kg/m^3^)	w/b
LY	440	66	44	656	602	-	1.7	143	0.29
LYB0.5	440	66	44	656	602	0.5	1.7	143	0.29
LYB1	440	66	44	656	602	1	1.7	143	0.29
LYB1.5	440	66	44	656	602	1.5	1.7	143	0.29
LYP0.5	440	66	44	656	602	0.5	1.7	143	0.29
LYP1	440	66	44	656	602	1	1.7	143	0.29
LYP1.5	440	66	44	656	602	1.5	1.7	143	0.29

Note: Super represents the superplasticizer.

**Table 5 materials-13-01715-t005:** Comparison of test and calculation results of *β_cu._*

Mixtures	*v*	$\sqrt{f_{t,f}/f_{t}}$	*β_cu,t_*	*β_cu,c_*	*β_cu,t_*/*β_cu,c_*
LY	0	0	1	1	1.00
LYB0.5	0.5	22.774	1.055	1.046	1.01
LYB1	1	22.774	1.115	1.091	1.02
LYB1.5	1.5	22.774	1.141	1.137	1.00
LYP0.5	0.5	14.404	1.008	1.029	0.98
LYP1	1	14.404	1.052	1.058	0.99
LYP1.5	1.5	14.404	1.094	1.086	1.01
0.3C [23]	0.3	29.193	1.027	1.035	0.99
0.6C [23]	0.6	29.193	1.057	1.070	0.99
0.9C [23]	0.9	29.193	1.122	1.105	1.02
PF0.6 [46]	0.6	8.54	1.027	1.020	1.01
PF0.9 [46]	0.9	8.54	1.034	1.031	1.00
PF1.2 [46]	1.2	8.54	1.052	1.041	1.01

**Table 6 materials-13-01715-t006:** Comparison of test and calculation results of *β_st._*

Mixtures	*v*	$\sqrt{f_{t,f}/f_{st}}$	*β_st,t_*	*β_st,c_*	*β_st,t_*/*β_st,c_*
LY	0	0	1	1	1.00
LYB0.5	0.5	22.680	1.108	1.102	1.01
LYB1	1	22.680	1.205	1.204	1.00
LYB1.5	1.5	22.680	1.282	1.306	0.98
LYP0.5	0.5	14.344	1.077	1.064	1.01
LYP1	1	14.344	1.151	1.129	1.02
LYP1.5	1.5	14.344	1.228	1.194	1.03
0.3C [23]	0.3	29.193	1.123	1.063	1.06
0.6C [23]	0.6	29.193	1.200	1.125	1.07
0.9C [23]	0.9	29.193	1.323	1.188	1.11
FG [47]	0.5	15.182	1.091	1.068	1.02
FP [47]	0.5	15.528	1.112	1.070	1.04
SF [47]	0.5	20.628	1.178	1.093	1.08

**Table 7 materials-13-01715-t007:** Comparison of test and calculation results of *β_f._*

Mixtures	*v*	$\sqrt{f_{t,f}/f_{f}}$	*β_f,t_*	*β_f,c_*	*β_f,t_*/*β_f,c_*
LY	0	0	1	1	1.00
LYB0.5	0.5	23.629	1.268	1.257	1.01
LYB1	1	23.629	1.412	1.372	1.03
LYB1.5	1.5	23.629	1.499	1.487	1.01
LYP0.5	0.5	14.944	1.172	1.205	0.97
LYP1	1	14.944	1.310	1.320	0.99
LYP1.5	1.5	14.944	1.410	1.435	0.98
0.3C [23]	0.3	31.623	1.371	1.259	1.09
0.6C [23]	0.6	31.623	1.428	1.328	1.07
0.9C [23]	0.9	31.623	1.486	1.397	1.06
BFRC2 [48]	0.5	9.287	1.150	1.170	0.98
BFRC3 [48]	1	9.287	1.390	1.286	1.08
BFRC4 [48]	1.5	9.287	1.500	1.401	1.07
BFRC5 [48]	2	9.287	1.633	1.516	1.08

**Table 8 materials-13-01715-t008:** Comparison of test and calculation results of *β_s._*

Mixtures	*v*	*l*	*β_s,t_*	*β_s,c_*	*β_s,t_*/*β_s,c_*
LY	0	6	1	1	1.00
LYB0.5	0.5	6	1.120	1.107	1.01
LYB1	1	6	1.242	1.213	1.02
LYB1.5	1.5	6	1.300	1.319	0.98
LYP0.5	0.5	3	1.093	1.083	1.01
LYP1	1	3	1.156	1.165	0.99
LYP1.5	1.5	3	1.251	1.248	1.00
PF0.6 [46]	0.6	12	1.079	1.185	0.91
PF0.9 [46]	0.9	12	1.206	1.278	0.94
PF1.2 [46]	1.2	12	1.286	1.371	0.94
GFR-RPC [49]	1.5	13	1.600	1.487	1.07

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
