# Peer review of "Mechanical Properties of Chopped Basalt Fiber-Reinforced Lightweight Aggregate Concrete and Chopped Polyacrylonitrile Fiber Reinforced Lightweight Aggregate Concrete"

_materials, 2020, doi:10.3390/ma13071715_

Round 1

Reviewer 1 Report

Strengths:

- β (strength enhancement parameter) and the obtained tests results can be used in future practical applications;

- the results obtained are well explained and supported by the existing literature and practical experience;

- supplementary tests like  oven-dried density and water absorption improved the scientific soundness;

Weaknesses:

- title and introduction need to be improved; now, it seems that the article refers to a lightweight aggregate concrete with a mixt of 2 types of fibers (basalt AND polyacrylonitrile), while there are 2 types of concrete, each of them having different fibers;

- using to many parameters for concrete recipes, like supplementary cementious materials (silica fume and fly ash) and a superplasticizer can reduce the applicability of the obtained results due to the difficulty of making the same mixture in different other situations / places; also, it is not explained the choosing of 12% for fly ash and 8% for silica fume;

- more, using a registered trademark product (Lytag lightweight aggregate) can limit the applicability of the study to some countries;

- the names of the LWAC mixtures should be detailed: LYB0.5 = LWAC + 0.5 volume of BF, etc;

- table 4, PCA parameter should be explained;

- figure 2 should be resized so that the text can be more easily read;

- line 148, “may because…” should be “may be because…”

- line 155, a reference to Part 4.1, which follows the present paragraph, should be replace with a short explanation;

- for all obtained results for mechanical characteristics, the values should be shown in a table or, at least, on the graphics (figures 6, 7, 8, etc); the lack of those values makes difficult to check the β parameters; for example, “when the volume fraction increased to 1% compressive strength achieved 63.7 MPa”, but when the volume fraction is 0.5, there is no value, only an improvement of 5% than plain LWAC;

- line 231, replace Romuldi with Romualdi; in the references list, Romualdi is [42] not [41]… all the references should be re-checked;

- figure 9 is missing;

- figures 10 and 11 should be rearranged in page and (b) should be replaced by (c) to preserve the logical order of the presentation (BF first and PANF second);

- αcu (strength correction coefficient) has different values (0.004 for compressive strength, 0.009 for splitting tensile strength, etc); the coefficient and its values are not explained;

- in all tables, parameter ft,f/t for LY should be zero;

- in table 5, all values for β should be verified; for example, according to the numbers there, for LYB1.5, 0.004×1.5×22.774+1= 1.136 not 1.147… this makes all the results untrusted! also, check the other values from all the tables;

- equation 7 – a new coefficient appears: 0.23… it is not explained;

- line 333, “by controlling growth…”; should add “of cracks”;

- equation 9 should be explained due to the new parameters included (0.117, 0.016); 

- the conclusions are mostly a reinterpretation of the obtained results, already presented in their paragraphs; only a few of them can be considered as conclusions (ex., “a 1.5% volume fraction of BF or PANF is suggested to obtain the optimal improvement in mechanical properties”); some possible practical applications can be mentioned;

- most references (2/3) are from Chinese authors/papers; a widely spread range of references can improve the quality of the present study and make the results trusted (for example, there are studies that recommend a volume fraction of 1% for better mechanical characteristics, not 1.5… of course, it depend of concrete and fiber type, but they can be a comparation borne);

- use superscript for 3 in m3 (line 19, 104, 117, etc);

- minor grammar and spelling mistakes – recheck the entire paper.

Author Response

Author's Response to Reviewer's Comments

Manuscript ID: materials-760008

List of changes

The changes in revised paper are marked in green.

Change 1: In revised paper, “A polycarboxylate high-performance water-reducing agent” have been modified to “A superplasticizer (high-performance water-reducing agent)”. See line 119.

Change 2: In Table 4, the “PCA” has been replace by “Super”, which represent the superplasticizer.

Change 3: In revised paper,“have been proven”have been modified to “have been proved” in line 61.

Change 4: In revised paper,“cementious”has been replaced by “cementitous” in line 117.

Change 5: The order of some references have been changed. Please see line 199 and 120

Thank you for your review and suggestions, we have revised our paper in accordance with your comments. The revised portions related to technical comments are marked in blue in the revised paper.

Responses to Reviewer #1:

Dear Reviewer #1,

Thanks for your comment and suggestion, and the author’s respond is shown as follow.

Comment 1: A title and introduction need to be improved; now, it seems that the article refers to a lightweight aggregate concrete with a mixt of 2 types of fibers (basalt AND polyacrylonitrile), while there are 2 types of concrete, each of them having different fibers.

Response 1: Thanks for reviewer’s comment. The title is changed to: Mechanical properties and design models of chopped basalt fiber-reinforced lightweight aggregate concrete and chopped polyacrylonitrile fiber reinforced lightweight aggregate concrete. The introduction is improved partially. In line 56-59, the sentences are modified to “Numerous studies showed the steel fiber significant effects on mechanical properties improvement. While, some defects of steel fiber must always be neglected, such as reducing workability, increasing weight of the concrete (because of the high specific gravity) and becoming perishable under water and salt solutions”. In line 62, “positive effects” is changed to “enhanced”. In line 72, the sentence is modified to “the behaviors of LWAC under compressive, splitting tensile strength and flexural loading”. In line 74, the word “fiber” is modified to “chopped fiber”. In line 83, a sentence “Moreover, existing literatures of fiber-reinforced LWACs lack the quantifying of the improved effects of fibers on strengths, especially on shear strength, which will limit the application.” is added. In line 85, the sentence is modified to “in this study, basalt fiber and polyacrylonitrile fiber are used to product LWAC specimens respectively.”

Comment 2: using to many parameters for concrete recipes, like supplementary cementious materials (silica fume and fly ash) and a superplasticizer can reduce the applicability of the obtained results due to the difficulty of making the same mixture in different other situations / places; also, it is not explained the choosing of 12% for fly ash and 8% for silica fume.

Response 2: Thanks for reviewer’s comment. Superplasticizer is one of water reducing agent, and many kinds of water reducing agents are used by different scholars. Even the same type of water reducing agents from different manufacturers also have some differences in chemical composition. Therefore, a kind of water reducing agent- superplasticizer used in this study can be viewed as a reference for readers, especially the dosage. The dosage of water reducing agent is determined by many times of trial mixture, and our research team have already used this water reducing agent to product ultra-high performance concrete, normal concrete and lightweight aggregate concrete. As for silica fume and fly ash, they have been widely used in concrete to make concrete environmental-friendly and improve concrete performance in strength, workability and durability. And the dosage of the silica fume and fly ash is determined by Wu [23], and we cite the reference in corresponding position at line 117

Comment 3: more, using a registered trademark product (Lytag lightweight aggregate) can limit the applicability of the study to some countries.

Response 3: Thanks for reviewer’s comment. We choose Lytag as a lightweight aggregate because Lytag is one of the industrial waste lightweight aggregates, and it has a mass production in China due to its high strength, as well as environmental protection behavior which followed the strategic objectives of sustainable development and green environmental protection. And literature “Georghiou, L.;Metcalfe, J. S.; Gibbons, M.; et al. Lytag: Light-Weight Aggregate from Pulverised Fuel Ash[M]// Post-Innovation Performance. Palgrave Macmillan UK, 1986.”showed that since 1982, Lytag had been manufactured by BORAL group of companies. Moreover, “Adell, V. Production of light weight aggregate from problematic industrial waste ashes using the lytag process. Imperial College London. 2007, April 24.” showed that Lytag had been applied in constructional engineering. Maybe it’s not widely used in other countries, but other kind of lightweight aggregate concrete can replace Lytag with the same size. Although there are some differences in chemical properties of them, but we believe that the macro performances of them are similar, so this study can be viewed as a reference to further researches.

Comment 4: the names of the LWAC mixtures should be detailed: LYB0.5 = LWAC + 0.5 volume of BF, etc.

Response 4: Thanks for reviewer’s comment. The names of LWAC mixtures are detailed in line 118-121: It is should be noted that, the detail of mixture names are explained as follow: “LY” represents Lytag lightweight aggregate; “B” and “P” represent BF and PANF respectively; number 0.5, 1 and 1.5 represent fiber volume fraction of 0.5%, 1% and 1.5% respectively.

Comment 5: table 4, PCA parameter should be explained.

Response 5: Thanks for reviewer’s comment. In Table 4, the value in eighth column represent the dosage of superplasticizer (high-performance water-reducing agent), therefore, word “PCA” has been replaced by “Super”, and in line 123, the explain of “Super” has been added: “Note: Super represent the superplasticizer.” See Change 2.

Comment 6: figure 2 should be resized so that the text can be more easily read.

Response 6: Thanks for reviewer’s comment. Figure 2 has been magnified properly.

Comment 7: line 148, “may because…” should be “may be because…”

Response 7: Thanks for reviewer’s comment. “may because” have been modified to “may be because” in line 154.

Comment 8: line 155, a reference to Part 4.1, which follows the present paragraph, should be replace with a short explanation.

Response 8: Thanks for reviewer’s comment. We delate the sentence “showing a similar development of the fiber effect on compressive strength as that mentioned in Part 4.1”, and giving a short explanation in line 162: “Based on the definition of specific strength, this increase phenomenon indicates that fiber play a more important role in crack-resistance than in filling in concrete.”.

Comment 9: for all obtained results for mechanical characteristics, the values should be shown in a table or, at least, on the graphics (figures 6, 7, 8, etc); the lack of those values makes difficult to check the β parameters; for example, “when the volume fraction increased to 1% compressive strength achieved 63.7 MPa”, but when the volume fraction is 0.5, there is no value, only an improvement of 5% than plain LWAC;

Response 9: Thanks for reviewer’s comment. In revised paper, An Appendix A has been built to list the average test values of strengths. Please see line 392.

Comment 10: line 231, replace Romuldi with Romualdi; in the references list, Romualdi is [42] not [41]… all the references should be re-checked.

Response 10: Thanks for reviewer’s comment. Romuldi has been replaced by Romualdi, and the reference number has been changed to [42]. Please see line 240.

Comment 11: figure 9 is missing.

Response 11: Thanks for reviewer’s comment. In line 256, “Figure 8” have been modified to “Figure 9”.

Comment 12: figures 10 and 11 should be rearranged in page and (b) should be replaced by (c) to preserve the logical order of the presentation (BF first and PANF second).

Response 12: Thanks for reviewer’s comment. Figure 10 and 11 have been rearranged in revised paper, and (b) has been replaced by (c) to preserve the logical order of the presentation. Please see line 259-261 and line 275.

Comment 13: αcu (strength correction coefficient) has different values (0.004 for compressive strength, 0.009 for splitting tensile strength, etc); the coefficient and its values are not explained.

Response 13: Thanks for the reviewer’s comment. In this study, the strength correction coefficients of different strengths have the same physical interpretation, that is the correction of fitting error. In this part, we discuss the fiber effects on strength enhancement, but in actual condition, the changes of strength aren’t only impacted by fibers, there are many factors could impact the test results. During the fitting process, we proposed this parameter to reduce the error. Therefore, for different strength testing, the strength correction coefficients are different. In order to make it clearer, a short sentence has been added to explain the effect of strength correction coefficients “(used to reduce the error and similarly hereinafter)”. Please see line 301.

Comment 14: in all tables, parameter ft,f/t for LY should be zero.

Response 14: Thanks for reviewer’s comment. In revised paper, the parameter of  in Tables 5, 6and 7 has been changed to be zero.

Comment 15: in table 5, all values for β should be verified; for example, according to the numbers there, for LYB1.5, 0.004×1.5×22.774+1= 1.136 not 1.147… this makes all the results untrusted! also, check the other values from all the tables.

Response 15: Thanks for reviewer’s comment. And all the values for β have been verified and the wrong values have been changed.

Comment 16: equation 7 – a new coefficient appears: 0.23… it is not explained.

Response 16: Thanks for reviewer’s comment. In revised paper, the explanation of coefficient 0.23 has been added in line 331: “it should be noted that the constant 0.23, is the correction coefficient of effect of v on flexural strength enhancement”

Comment 17: line 333, “by controlling growth…”; should add “of cracks”.

Response 17: Thanks for reviewer’s comment. In revised paper, “of cracks” have been added in line 348.

Comment 18: equation 9 should be explained due to the new parameters included (0.117, 0.016).

Response 18: Thanks for reviewer’s comment. The explanation of new parameters have been added in line 350: “herein, constant 0.117 is the correction coefficient of effect of v on shear strength enhancement, and constant 0.016 is the correction coefficient of coupling effect of v and l on shear strength enhancement.”

Comment 19: the conclusions are mostly a reinterpretation of the obtained results, already presented in their paragraphs; only a few of them can be considered as conclusions (ex., “a 1.5% volume fraction of BF or PANF is suggested to obtain the optimal improvement in mechanical properties”); some possible practical applications can be mentioned;

Response 19: Thanks for reviewer’s comment. Conclusion has been improved, please see the line 169, 377 and 382

Comment 20: most references (2/3) are from Chinese authors/papers; a widely spread range of references can improve the quality of the present study and make the results trusted (for example, there are studies that recommend a volume fraction of 1% for better mechanical characteristics, not 1.5… of course, it depend of concrete and fiber type, but they can be a comparation borne).

Response 20: Thanks for reviewer’s comment. We have changed and added some references from other countries, please see the Reference [1], [16-18], [48-50].

Comment 21: use superscript for 3 in m3 (line 19, 104, 117, etc).

Response 21: Thanks for reviewer’s comment. In revised paper, all the “m3” have been modified to “m3”. Please see line 21, 102, 159, 152, 153, 161, 162 and Tables 1, 2, 3 and 4.

Comment 22: minor grammar and spelling mistakes – recheck the entire paper.

Response 22: Thanks for reviewer’s comment. In fact, we have sent this paper to have an English Editing Service already. But we also have rechecked the entire paper and have modified some grammar and spelling mistakes.

Reviewer 2 Report

Comments

This paper investigated the application of FRP in concrete. The outcome is interesting to readers. However, there are several aspects that need to be improved. The reviewer can only recommend for publication if the author satisfactorily address the following comments in the revised version.

  1. The pore space in concrete can be filled with matrix but not fibres. How the increase of fibre content reduced the water absorption for concrete? An explanation is needed.
  2. The enhancement parameters used to predict strength is purely empirical and did not consider the effect of fibre lengths and fibre diameters. It is important to describe how the fibre lengths and diameters can affect the properties of concrete?
  3. Is the proposed enhancement parameters are applicable for other types of fibre to predict the properties? This can be validated by using the data from the published literature where the researchers used different types of fibre.
  4. There are few recent applications of fibre reinforcement in concrete structures. For example, FRP retaining wall system [Ref: Short-term flexural behaviour of concrete filled pultruded GFRP cellular and tubular sections with pin-eye connections for modular retaining wall construction], fibre reinforced concrete slab [Ref. Flexural behaviour of concrete slabs reinforced with GFRP bars and hollow composite reinforcing systems] , column repairing using FRP [Experimental and numerical evaluations on the behaviour of structures repaired using prefabricated FRP composites jacket]. These recent application should be highlighted in the introduction section to improve the background study.

Author Response

Author's Response to Reviewer's Comments

Manuscript ID: materials-760008.

Responses to Reviewer #2:

Dear Reviewer #2,

Thanks for your comment and suggestion, we have revised our paper in accordance with your comments. The revised portions related to technical comments are marked in blue in the revised paper, and the author’s respond is shown as follow.

Comment 1: the pore space in concrete can be filled with matrix but not fibres. How the increase of fibre content reduced the water absorption for concrete? An explanation is needed.

Response 1: Thanks for reviewer’s comment. Liu [40] test results also showed that fiber addition into concrete could decrease the water absorption. And Karahan [41] reported that, there was a close relationship between the porosity and water absorption that water absorption value increases as the porosity value increases. Merely, some type of fibers could increase the porosity of concrete, example, as content of PP excess 1%, the porosity will increase by reference “Richardson, A. E.; Coventry, K. A.; Wilkinson, S. Freeze/thaw durability of concrete with synthetic fiber additions. Cold Regions Science and Technology. 2012, 83, 49–56.”

Therefore, in revised paper, we have explained the reason in line 175: “Liu [40] showed the similar test results. According to Karahan [41], the water absorption is closely related to porosity and water absorption would decrease as the porosity reduces.”

Comment 2: The enhancement parameters used to predict strength is purely empirical and did not consider the effect of fibre lengths and fibre diameters. It is important to describe how the fibre lengths and diameters can affect the properties of concrete?

Response 2: Thanks for reviewer’s comment. As for cube compressive strength and splitting tensile strength of LWACs, we had considered the lengths and diameters, but the fitting results were not good since we had applied the thoughts and methods mentioned in part 4.1 and part 4.2, so they are neglected. As for flexural strength and shear strength of LWACs, we have considered fiber lengths, but do not use diameters, because the value of diameters are very small compared with length and strength, if diameter is considered into Equation (7) and (9), it will has a much higher coefficient than other parameters which can decrease the impacts by other parameters, in other words, changing the value of other parameters, the calculation results will show almost no differences. Another reason is that, the fitting results are not good as using fiber diameters. Therefore, as for compressive strength and splitting tensile strength, the length and diameter of fibers are not considered, and as for flexural strength and shear strength, diameter of fibers are not considered.

Comment 3: Is the proposed enhancement parameters are applicable for other types of fibre to predict the properties? This can be validated by using the data from the published literature where the researchers used different types of fibre.

Response 3: Thanks for reviewer’s comment. In revised paper, the data form the published literatures have been referenced to validate the enhancement parameters proposed in this study. Please see Tables 5, 6, 7 and 8 and Reference [26], [47-50]. And the corresponding brief descriptions have been added in line 305, 320, 336 and 355.

Comment 4: There are few recent applications of fibre reinforcement in concrete structures. For example, FRP retaining wall system [Ref: Short-term flexural behaviour of concrete filled pultruded GFRP cellular and tubular sections with pin-eye connections for modular retaining wall construction], fibre reinforced concrete slab [Ref. Flexural behaviour of concrete slabs reinforced with GFRP bars and hollow composite reinforcing systems] , column repairing using FRP [Experimental and numerical evaluations on the behaviour of structures repaired using prefabricated FRP composites jacket]. These recent application should be highlighted in the introduction section to improve the background study.

Response 4: Thanks for reviewer’s comment. In line 49, the reference [Short-term flexural behaviour of concrete filled pultruded GFRP cellular and tubular sections with pin-eye connections for modular retaining wall construction], [Flexural behaviour of concrete slabs reinforced with GFRP bars and hollow composite reinforcing systems] and [Experimental and numerical evaluations on the behaviour of structures repaired using prefabricated FRP composites jacket] have been cited, please Reference 16-18.

Reviewer 3 Report

The problem of fiber reinforcing concrete presented in this manuscript is very current. The article is carefully written and complies with the requirements of Mdpi.

Drawings could be on a slightly larger scale - mainly descriptions of photos in figure 2. The composition of figures 10 and 11 is also incorrect (maybe a print error to pdf).

I will start my substantive review from the title. I can't agree with the authors that they presented any design models of lightweight aggregate. Based on the statistical analysis of the test results, they only proposed formulas to determine the basic strength properties, and this is far from design model. I think it would be more appropriate to limit the title to "Mechanical properties of lightweight aggregate concrete with chopped basalt and polyacrylonitrile fiber reinforcement”.

There is no information as to how the properties of the materials presented in Tables 1-3 have been determined, whether on the basis of the authors' own research, or are these data obtained from the manufacturer of these materials?

Research procedures are clearly explained, however, here my remark concerns the research methods applied. A four-point bending test was used to determine the flexural strength. I understand that this test is sufficient to determine the strength alone, but in my opinion in the case of fiber reinforced concrete it is very important to describe the “post peak” behavior. This possibility is provided by three-point bending of the notched beam, in accordance with the Rilem recommendation TC 162-TDF. This test is currently the world standard. In my opinion, the authors in the article should at least explain why this test was not carried out. How should the design of FRC elements look like when we know only strength? Should linear-elastic models be used?

The biggest weakness of the tests described in the manuscript is the lack of deformation measurements. Nowadays, technology offers enormous possibilities, for example Digital Image Correlation. This would allow authors to understand the destruction process much better. It would also enable the description mentioned above “post peak behaviour”. This is my suggestion for future research.

Author Response

Author's Response to Reviewer's Comments

Manuscript ID: materials-760008

Responses to Reviewer #3:

Dear Reviewer #3,

Thanks for your comment and suggestion, we have revised our paper in accordance with your comments. The revised portions related to technical comments are marked in blue in the revised paper, and the author’s respond is shown as follow.

Comment 1: Drawings could be on a slightly larger scale-mainly descriptions of photos in figure 2. The composition of figures 10 and 11 is also incorrect (maybe a print error to pdf).

Response 1: Thanks for reviewer’s comment. In revised paper, Figure 2 has been magnified properly. And Figure 10 and 11 have been rearranged.

Comment 2: I will start my substantive review from the title. I can't agree with the authors that they presented any design models of lightweight aggregate. Based on the statistical analysis of the test results, they only proposed formulas to determine the basic strength properties, and this is far from design model. I think it would be more appropriate to limit the title to "Mechanical properties of lightweight aggregate concrete with chopped basalt and polyacrylonitrile fiber reinforcement”.

Response 2: Thanks for reviewer’s comment. The title has been modified to be “Mechanical properties and design models of chopped basalt fiber-reinforced lightweight aggregate concrete and chopped polyacrylonitrile fiber reinforced lightweight aggregate concrete”

Comment 3: There is no information as to how the properties of the materials presented in Tables 1-3 have been determined, whether on the basis of the authors' own research, or are these data obtained from the manufacturer of these materials?

Response 3: Thanks for reviewer’s comment. The properties of materials presented in Tables 1-3 are determined from the manufacturers, and we have added this information in revised paper that “It also should be note that all the properties presented in Tables 1-3 are determined by manufacturers.” Please see line 103.

Comment 4: Research procedures are clearly explained, however, here my remark concerns the research methods applied. A four-point bending test was used to determine the flexural strength. I understand that this test is sufficient to determine the strength alone, but in my opinion in the case of fiber reinforced concrete it is very important to describe the “post peak” behavior. This possibility is provided by three-point bending of the notched beam, in accordance with the Rilem recommendation TC 162-TDF. This test is currently the world standard. In my opinion, the authors in the article should at least explain why this test was not carried out. How should the design of FRC elements look like when we know only strength? Should linear-elastic models be used?

Response 4: Thanks for reviewer’s comment. It no doubt that post-peak behavior is an important property. Actually, during the test, the stress-strain curves are measured as well, including axial compressive test, splitting tensile test, flexural test and shear test (In Figure 2 the displacement and strain gauge can be found). Therefore, we have the data about the post-peak behavior of LWACs. But in this paper, we only focus on strength behavior by fiber enhancement effect. And the post-peak behavior of LWACs are under working.

Comment 5: The biggest weakness of the tests described in the manuscript is the lack of deformation measurements. Nowadays, technology offers enormous possibilities, for example Digital Image Correlation. This would allow authors to understand the destruction process much better. It would also enable the description mentioned above “post peak behaviour”. This is my suggestion for future research.

Response 5: Thanks for reviewer’s comment. As we mentioned above, we have obtained the post-peak behavior of LWACs during all the test in this paper, but we only discuss the strength performance of LWAC samples. Our research group is working on the post peak behavior of fiber reinforced lightweight aggregate concrete under different types of loading.

Round 2

Reviewer 1 Report

The authors responded successfully to the majority of previous comments. Although there are still some uncertainties regarding the β coefficient and its way of calculation, adding some results from already published papers is a good start for future researches.

Continue your work because the idea is good and can be very useful for many people involved in this sector.

Reviewer 3 Report

I want to thank the authors for all the corrections. Of course, it is a pity that the article did not include research covering post peak behavior. However, I understand the argument that they are still being processed and I hope they will be published soon.